# A Quantitative Positive Energy District Definition with Contextual Targets

**Simon Schneider** [1,*] **, Thomas Zelger** [1] **, David Sengl** [1] **and José Baptista** [2]

1   Department of Industrial Engineering, University of Applied Sciences Technikum Wien, 1200 Vienna, Austria
2   Department of Engineering, School of Science and Technology, University of Trás-os-Montes and Alto Douro, INESC-TEC, UTAD's Pole, 5000-801 Vila Real, Portugal
*   Correspondence: simon.schneider@technikum-wien.at

**Abstract:** This paper presents the goals and components of a quantitative energy balance assessment framework to define Positive Energy Districts (PEDs) flexibly in three important contexts: the context of the district's density and local renewable energy supply (RES) potential, the context of a district's location and induced mobility, and the context of the district's future environment and its decarbonized energy demand or supply. It starts by introducing the practical goals of this definition approach: achievable, yet sufficiently ambitious, to be inline with Paris 2050 for most urban and rural Austrian district typologies. It goes on to identify the main design parts of the definition—system boundaries, balancing weights, and balance targets—and argues how they can be linked to the definition goals in detail. In particular, we specify three levels of system boundaries and argue their individual necessity: operation, mobility, and embodied energy and emissions. It argues that all three pillars of PEDs, energy efficiency, onsite renewables, and energy flexibility, can be assessed with the single metric of a primary energy balance when using carefully designed, time-dependent conversion factors. Finally, it is discussed how balance targets can be interpreted as information and requirements from the surrounding energy system, which we identify as a "context factor". Three examples of such context factors, each corresponding to the balance target of one of the previously defined system boundaries, operation, mobility, and embodied emissions, are presented: density (as a context for operation), sectoral energy balances and location (as a context for mobility), and an outlook on personal emission budgets (as a context for embodied emissions). Finally, the proposed definition framework is applied to seven distinct district typologies in Austria and discussed in terms of its design goals.

**Keywords:** Positive Energy District; PED definition; context factors; PED assessment; energy transition; energy balance assessment; sustainable districts; key performance indicators

## 1. Introduction

The need for a Positive Energy District (PED) definition stems from both EU and individual project levels: on the level of the European Union (EU), it is necessary to uniformly define PEDs to measure the success of the strategic SET Plan mission of bringing a hundred PEDs underway by 2025 [1], and at the same time individual projects need criteria that can be met to be referenced and possibly certified as PEDs. As a consequence, PED's definition has been discussed extensively [2–6], and although there is recognition of the need for a common definition, none could be arrived at as of yet [5–7]. Many approaches have been put forward that do not or do not exclusively employ a quantitative assessment scheme [8–10], but many researchers and practitioners agree that the definition of a PED must ultimately entail the evaluation of an energy balance, whose result must be positive [1,2,4–6,11].

The differences between existing definitions that employ such a positive energy balance as a sufficiency criterion can thus be outlined: firstly, the question of which energy

services should be considered. There exist many approaches [6], but most converge on the minimum for the operation of the district's heating, ventilation, and air-conditioning (HVAC) systems [9] and sometimes also user electricity such as plug loads [2,12]. Mobility and embodied energy, on the other hand, are less prevalent due to the apparent negative effects on a positive energy balance or the lack of suitable assessment methodologies in theory and practice. Other approaches forego a uniformly quantifiable definition altogether and instead make this determination a project-specific process [2,3].

The second divergence is the balance metric or key performance indicators (KPI) to be used: energy end-use or flexibility KPIs [13], total or non-renewable primary energy [2,12,14], or greenhouse gas (GHG) emissions [15] or a combination of the above and others [8,13]. In the latest Annex 83 review of the International Energy Agency (IEA), however, most definitions use a primary energy indicator, with notable exceptions [16]. However, differences in primary energy conversion can cause drastically different balance assessments [17–19].

Thirdly, there is divergence in the kind of system boundaries the balance is evaluated on, although "the majority of PEDs in Europe apply the dynamic-PED concept, with geographical boundaries" [6] (p.13). This can be further obfuscated by the fact that many projects do not readily achieve their positive balance without some form of "offsite subsidies", be it in the form of RES credits or outright including these resources in the PED boundary [2,9], but not necessarily clear rules as to how this inclusion must be performed. Temporally, most use an annual balancing period of an operation year [16].

What is typically not discussed at length is the goal and scope of the definition: is it suitable for green or brown field developments, any climate zone, and density? Should it be possible for any district, and of which ambition level to achieve a "PED"? As with other European standardization processes such as the Energy Performance of Buildings directive (EPBD [20]), one could separate what a PED is in different levels of regionality from European, national, and municipal all the way down to project-specific. In practice, however, most definitions are developed in international or national projects but used mostly by the districts within that project. This further adds to the conundrum of which definitions can and should be used for which PED projects, and most importantly: why. In fact, using multiple different definitions on different levels of regionality and detail, from broad European frameworks through national standardizations to project-level specification could be a promising way forward. In the meantime the need to bridge the gap between a standardized and unambiguous definition for EU reporting on one side and the flexibility to account for local contexts and feasibility on the other has become abundantly clear [4,7]. This paper aims to add to the theory underpinning PED definitions by introducing a conceptual view on the positive energy balance, in which the assumed target of positivity can be calibrated by the use of so-called "context factors" (CF for short). These are virtual balance components that need to be designed district-independently to maintain PED feasibility in the desired context.

## 2. Methods and Approach: PED Definition as a Design Problem

It can be argued that creating a definition is in itself not a scientific process but rather a design problem [21]. A definition cannot be observed, theorized upon, and validated. Instead, it specifies and regulates the appropriate use of language. In this case, "appropriate" is very open for interpretation (compare e.g. [22]). If the definition is to be useful, it needs to show exactly how. For that reason, it is our understanding that the most important part of the PED definition is the argument as to **why** it was designed in any particular way, and it must be very articulate about the goals it is aimed at. Most definitions indeed solve many of the varying stakeholder problems, but they often do so implicitly rather than stating these plainly. Only if the aims of the definition are declared alongside, can it be tested if it is indeed suited to fulfil them. The following figure illustrates this taken design approach starting with the goals and deriving the criteria and their operationalization thereafter:

After the definition's goals, the actual components of a PED definition were identified as a quantitative balance assessment that can be structured in three main parts (in accordance with [21]): (1) system or balance boundaries, (2) a balance weighting system, and (3) a balance target, which we will proceed to construct by means of the aforementioned "context factors" (CF), which, as the name implies, are derived project-independently from a national PED context. This is illustrated in Figure 1.

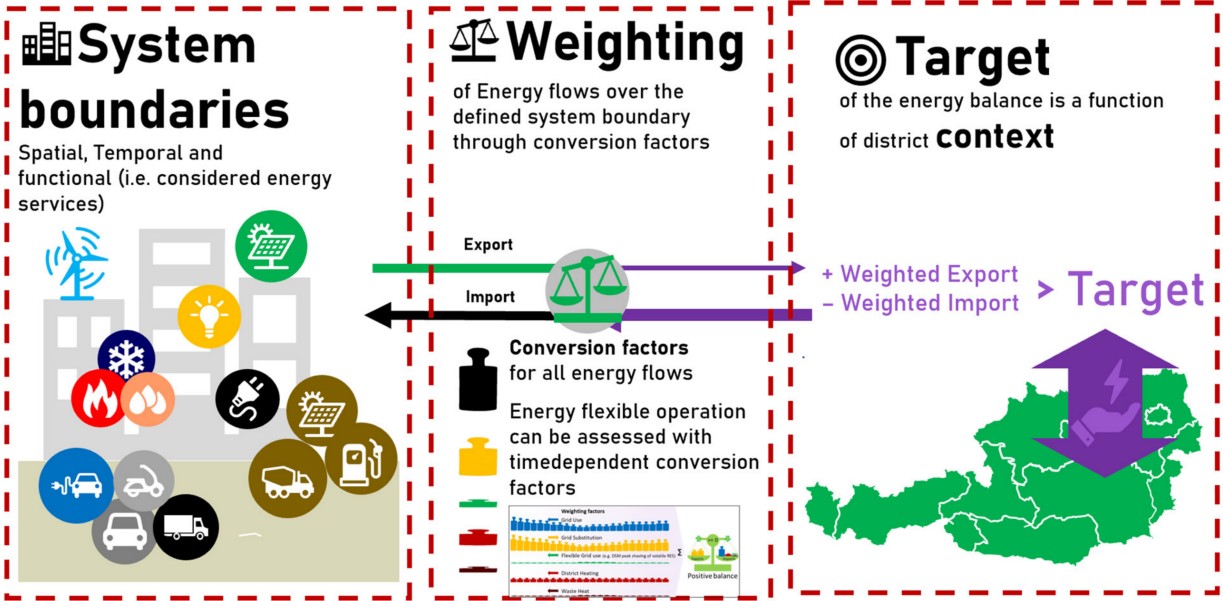

**Figure 1.** The three parts of a PED definition via an energy balance assessment.

These three areas of the district system boundary, system balance weighting, and balance targets correspond to the three questions a quantitative PED definition via an energy balance needs to address in unison as the definition design problem. From the start, the balance target is not necessarily positive or zero, but can, in principle, be a function of any set of parameters deemed relevant. Thus, the definition of an energy balance target can include both project-intrinsic and project-extrinsic factors. On the one hand, this is an additional challenge, but at the same time it is an opportunity: With this, dynamic external requirements can be related to project-specific proposed solutions.

This might be surprising, as it seems to contradict the prevailing understanding of a hard-set target of "greater than zero". However, it can be argued that targets can be reformulated as context factors and, as such, be included as virtual demands and supplies in a still formally positive balance. In fact, many current definitions already use some form of virtual demands and supplies to modulate the balance feasibility. Most notably, not including certain energy services in the balance, such as mobility and embodied energy, is equivalent to instead including a virtual supply of equal size from somewhere to offset the demand. The goal of this paper is to provide a formalism in which this somewhere can be defined more transparently and uniformly.

## 3. Goals of the PED Definition

The PED definition cannot be separated from its goals and implications in practice. The presented definition's goal is: to be achievable, yet sufficiently ambitious to be compatible with Paris 2050 for any urban and rural Austrian district typology. This is in line with the EU Commission's statement that the ambition of PEDs is to "go well beyond what is already requested in the Energy Performance of Buildings Directive" [23], but also aims at the sufficiency question as well. The definition development of the approach was therefore done under these guiding objectives:

1. The PED definition contains all relevant features of a future 100% renewable energy system. Such PEDs anticipate future requirements by the precautionary principle and must take into consideration their future surroundings.

2. The PED definition is achievable in both rural and urban contexts, or areas of low and high building density (technically, legally, and economically). Lower densities should not be implicitly favored by the PED definition.

3. The PED definition is achievable for different types of usage mixes with comparable ambition, not just for uses with low energy demand or good temporal alignment between supply and demand.

4. The PED definition's achievability is not dependent on the incidental but uncommon availability of local renewables such as local (industrial) waste heat, hydro, or wind power.

5. The PED definition links the national climate goals (i.e., a decarbonized future energy supply) with the local targets of a district in a comprehensive quantitative system.

6. The PED definition is compatible with the definition developed at the European level by the Alignment Task-Force JPI UE Framework Definition [1].

7. The PED definition has directional stability and consistency for all process phases: From project development to implementation or monitoring (zoning, architectural competition, planning, execution, and operation). This ultimately means a stable definition operationalization and accompanying assessment framework as part of a nationally accredited standardization and certification scheme.

8. The PED definition concept should be flexible and extendable, from operation (PED Alpha) to mobility (PED Beta) to the entire life cycle (PED Omega), and should lend itself to transparent reparameterization in the future.

In this paper, the introduction of a quantitative PED definition assessment scheme using context factors is aimed at facilitating these goals. In particular, the density context-factor addresses Goals 1–3 and 5 by connecting national PV capacity targets (Goal 1 and 5), depending on density (Goal 2), and is comparably achievable for different usages (Goal 3), as shown in Section 7. Goal 4 is facilitated by introducing other means of achieving a nominally positive energy balance. The proposed is also compatible with the JPI UE Framework definition (Goal 6), as the latter also mandates a positive energy balance but also expects some form of contextualization. This paper also tries to further Goal 7 by introducing a formal definition framework that can subsequently be parametrized for national standardization and certification. Goal 8 is addressed by the differentiation of the definition into three possible perspectives through the introduction of three possible system boundaries (Alpha, Beta, Omega).

The goal of this definition is to envisage the PED as part of a future decarbonized energy system and to quantitatively relate the PED balance targets to the achievement of these national and international climate goals. This requires a quantitative method of allocation and effort sharing, which was presented in [24] and will further be detailed in Sections 4–6.

The aim of the definition is to make the flexibility of the district quantifiable, such that higher energy flexibility simultaneously has a positive impact on the achievement of the PED balance target. This is important to quantitatively link all three energy dimensions of a PED, i.e., energy efficiency, local renewable generation, and energy flexibility. This also allows different districts to prioritize and realize their respective potentials for achieving the PED definition according to their local circumstances without the need for further definitional additions. This is addressed as part of the balance weighting system in Section 5.

### Non-Goals

The focus of the PED definition design is encapsulating the climate neutrality requirements of the built environment through the means of a positive energy balance. Therefore, an explicit design choice is made to not concern the definition with possible additional—albeit

important—dimensions and aspects of district development and assessment. Instead, the definition should be compatible and work in conjunction with already existing assessment and certification systems that consider these aspects. Additionally missing in this definition are criteria and specifications for the district development and planning process, as different stakeholders and ownership structures (such as new construction versus existing) and spatial and urban planning organizations require different planning processes. Prescribing a specific process or awarding points complicates and distracts from the main design goals of this PED definition. Instead, districts can use dedicated systems for this, such as the Austrian klima:aktiv standard for districts and neighborhoods [25], Leadership in Energy and Environmental Design (LEED [26]), or the Building Research Establishment Environmental Assessment Method (BREEAM [27]).

Another non-issue is the definition of important aspects of social, economic, and environmental dimensions. This is not to say that these aspects of district development such as social inclusion, safety and comfort, community development, and the creation of a sustainable ecosystem are not equally, if not more, important than the energy assessment. However, they should be better covered by assessment and certification systems focused on them and complementary to achieving the PED definition, such as Total Quality Building (TQB), klima:aktiv [28], local requirements of subsidized housing, and other regulatory and legal instruments [29].

This approach also foregoes the definition of minimum or sufficiency criteria for individual aspects such as energy efficiency, renewable production, and flexibility: Because districts have very diverse typologies, the definition of accurate individual criteria is correspondingly complex. Sufficiency criteria, e.g., defined in the building code or other standards, are helpful but not necessary in the context of a PED definition: A district should be free to decide *how* to best use its potential and how it contributes to climate neutrality. Complementary systems could, again, be passive house standards [30] and the local building code [31], instruments of zoning and development plans, urban planning framework agreements [32], etc.

## 4. System Boundaries

System boundaries are considered in a spatial, temporal, and functional sense in accordance with the fundamentals of PED energy modeling described in [11]. The definition of system boundaries is required to enable the balancing of flows over these defined boundaries. As such, it is necessary to distinguish and define all three different types:

Spatial means an actual physical boundary of included energy services and supplies. In other approaches, this boundary is sometimes used to make "offsite" RES party possible. In the presented approach, the available "offsite RES" is instead addressed as part of every PED's surrounding in the form of a balance-target-adjusting context factor, introduced in Section 6 and not subject to spatial boundaries.

Temporal system boundaries can be interpreted as the balancing period and are typically set to one operational year.

Functional system boundaries are used to identify specific energy functions, uses, or demands to be included or excluded according to function, rather than spatial proximity. Functional system boundaries can be further differentiated into renewable energy supply within the system boundary, referred to as "onsite", and energy services to be accounted for in the balance. Note that "onsite" here does not necessarily mean spatially onsite but rather "within the system boundary", counting positively towards the energy balance.

The functional system boundaries and the included energy services can be roughly grouped into three regimes of increased responsibility: (1) operational energy and user electricity, (2) mobility, and (3) embodied energy and emissions. This approach defines three variants, or shells, from PED Alpha in the innermost part considering only the operating energy, over PED Beta including private everyday mobility, up to PED Omega in the outermost part, where the embodied energy of district construction, maintenance, repair, and mobility are also considered. The three system boundaries reflect three areas

of quantitative assessment: operation, mobility induced by the location, and embodied Energy, respectively. Each is associated with increasing effort and greater uncertainty than the last. The expandability of the system boundary is important because, on the one hand, different data and information are available in the course of the project, and on the other hand, there are already partly considerable differences between projects in the available data and the objectives. At the same time, appropriate data are necessary for simulation and verification. The system boundaries are illustrated in Figure 2 and are detailed in the following subsections.

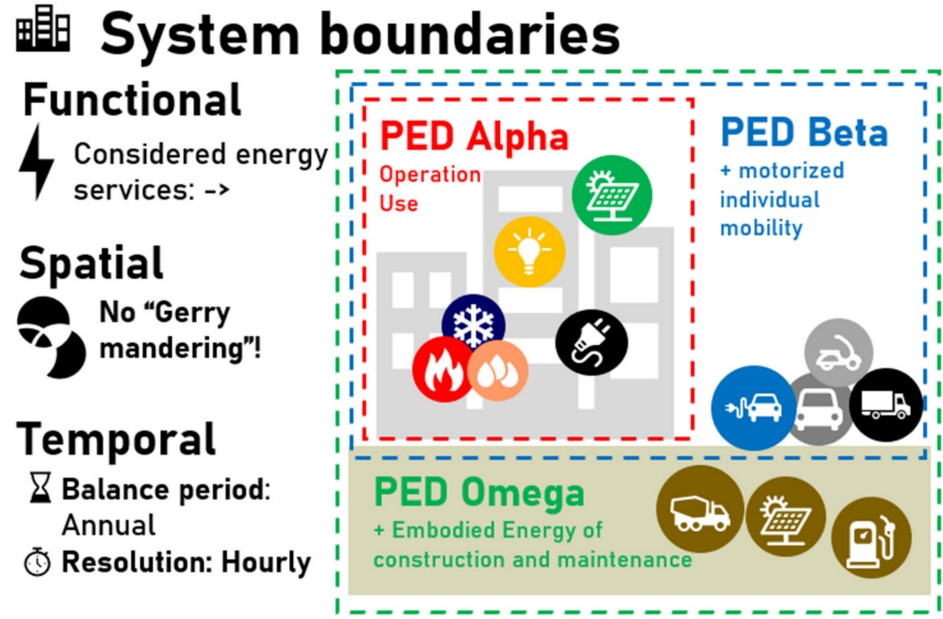

**Figure 2.** Types and extent of defined system boundaries.

### 4.1. Functional System Boundary: Considered Energy Services

The definition considers all energy demand for building operation, domestic hot water, lighting, and building services as well as the energy demand for living, working, and services (e.g., appliances, computers). Process energy is not considered directly, but indirectly through the crediting of national surpluses or deficits. Everyday individual mobility and embodied energy are also included in their respective boundaries, as listed in Table 1.

### 4.2. Spatial System Boundary

The spatial system boundary coincides with the physical district boundary. All definition components refer to the area that is necessary for the full use of the district. A "Gerrymandering" of the spatial boundaries for targeted inclusion or exclusion for energy and emission balance reasons must be avoided. District boundaries should be as convex as possible and as concave as necessary. Within these spatial boundaries, all local renewable energy sources are permissible regardless of the conversion technology and can be used to cover the energy balance as long as the assumption holds that its use within the district does not limit the usability for areas outside the district. This is particularly important to consider for the thermal and electrical use of flowing water for cooling and small hydropower. Further than that, it does not seem practical to generalize the validity and shape: The spatial system boundary should be drawn in such a way that the immediately surrounding areas do not suffer any obvious disadvantage from becoming PEDs themselves.

**Table 1.** Considered energy services within PED boundaries Alpha, Beta, and Omega.

| Energy Services | | Alpha | Beta | Omega | Implicit * |
|---|---|:---:|:---:|:---:|:---:|
| **Building operation** | Heating | ✓ | ✓ | ✓ | - |
| | Cooling | ✓ | ✓ | ✓ | - |
| | Humidification and dehumidification | ✓ | ✓ | ✓ | - |
| | Ventilation | ✓ | ✓ | ✓ | - |
| | Auxiliary power of the building services system | ✓ | ✓ | ✓ | - |
| | General power and lift | ✓ | ✓ | ✓ | - |
| | Lighting | ✓ | ✓ | ✓ | - |
| **District operation Industry, agriculture** | Power requirements of users (plug loads) | ✓ | ✓ | ✓ | - |
| | Operating power (office, retail, school) | ✓ | ✓ | ✓ | - |
| | Process heat | - | - | - | ✓ |
| | Process cooling | - | - | - | ✓ |
| | Electricity demand for industrial production processes | - | - | - | ✓ |
| | Electricity demand for general use (incl. services) | ✓ | ✓ | ✓ | - |
| **Mobility** | Motorized private transport | - | ✓ | ✓ | - |
| | Public transport | - | - | - | ✓ |
| | Other mobility | - | - | - | - |
| **Embodied Energy** | Components of the Austrian energy certificate | - | - | ✓ | - |
| | Accessory components (cellars, underground parking, garages, carports, bicycle storage areas, balconies and terraces, other outbuildings) | - | - | ✓ | - |
| | Building and energy equipment | - | - | ✓ | - |
| | Vehicles and infrastructure for mobility | - | - | ✓ | - |
| | Public transport | - | - | - | ✓ |

(✓) Included in system boundary, (-) not included in system boundary. * **Implicit**: Not included in any system boundary but instead part of the district context of the surrounding energy scenario that in turn influences the balance target as context factors.

## 5. Balance Weighting System

The weighting and the associated evaluation of energy flows is a central and controversially discussed topic of the PED definition. The approach presented here focuses less on the physical self-sufficiency or autonomy of the district but rather on the assessment of the district's contribution to the climate neutrality of the overarching national energy system. Specific weighting objectives therefore are:

1. Linking to planning practice and existing literature: The use of total primary energy and GHG emissions by means of conversion factors from the current building code or, in the case of district heating, county-specific regulations.
2. Mapping of seasonal differences: Monthly conversion factors based on Austrian building codes [31], renewable feed-in during summer and grid import in winter are weighted differently due to their different grid support and substitution alternatives.
3. The evaluation of energy flexible, grid-serving, i.e., time-sensitive, grid use and feed-in: Otherwise unavailable energy in the surrounding system is weighted with a conversion factor of zero.
4. Biomass use is possible, but not implicitly preferred due to low conversion factors in the building code: Instead, an average of total and non-renewable primary energy is used. If only the first were used, biomass would mostly be infeasible, and if only the latter were used, biomass systems would easily outperform electricity-based systems. These goals led to the conversion factors presented in Table 2, which are further discussed in Section 8.

### 5.1. Energy-Flexible Grid Use through Demand-Side Management (DSM)

Although other indicators such as the Grid Support Coefficient [33] or the Smart Readiness Indicator [34–36] facilitate more in-depth assessments of energy flexibility and

grid serviceability, they can also increase the complexity of modeling and verification. Instead, this PED definition aims to assess a district's capabilities for energy-flexible and grid-serving operation through a time-dependent weighting of the energy flows instead of mandating an additional KPI. The resulting energy balance reflects energy efficiency, local renewable generation, and energy flexibility measures. This also allows different districts different approaches and strategies in achieving a positive energy balance.

**Table 2.** Weighting factors for energy flows over the defined system boundaries.

| Energy Flow | PED Alpha, PED Beta | PED Omega | Source |
|---|---|---|---|
| Uncontrolled grid use and feed-in | Total Primary energy Monthly conversion feed-in sign-inversed | $CO_2$-equiv. Monthly conversion feed-in sign-inversed | National building code [31] |
| Energy-flexible grid use (DSM) | Zero | Zero | Section 5.1 |
| Biomass | 100% renewable + 50% non-renewable primary energy | $CO_2$-equiv. | |
| Other energy carriers | Total primary energy | $CO_2$-equiv. | National building code [31] |
| Fuels (mobility) | Total primary energy | $CO_2$-equiv. | |

In operationalizing the time-dependence of the weighting factors, a distinction is made between "generally available regional renewables" and "situational RES", which are useable only by an appropriately energy-flexible district—i.e., the energy flexibility provided by the district. Only the latter is considered favorably in the balance. The former is not disregarded but handled as a context factor, as shown in Section 6. The flexibility is assessed by weighting with a primary energy conversion factor of zero in the balance, but is subject to operational constraints: This temporally available renewable electricity is situational and would not be useable without corresponding strategies and regulations. Enabling this integration is a key goal of energy flexibility PEDs and is therefore treated differently in terms of methodology in principle. For a 100% renewable Austria, expansion rates of wind power by a factor of 5 and PV by a factor of 20 are needed, depending on the scenario, which means that strong seasonal fluctuations have to be compensated.

How can this energy flexibility be provided? The physical principles presented in IEA EBC Annex 67 on energy-flexible buildings [37] (p.67) apply: In particular, the thermal storage masses of buildings and districts with good thermal insulation and heavy building components can lead to significant displacement periods of several days, during which the system can freely choose the timing of its energy use for heating and cooling [38]. The principle is schematically summarized in the following Figure 3. The DSM operation is included and assessed in the energy balance by lowering the associated weighting of energy import during times of regionally available surpluses.

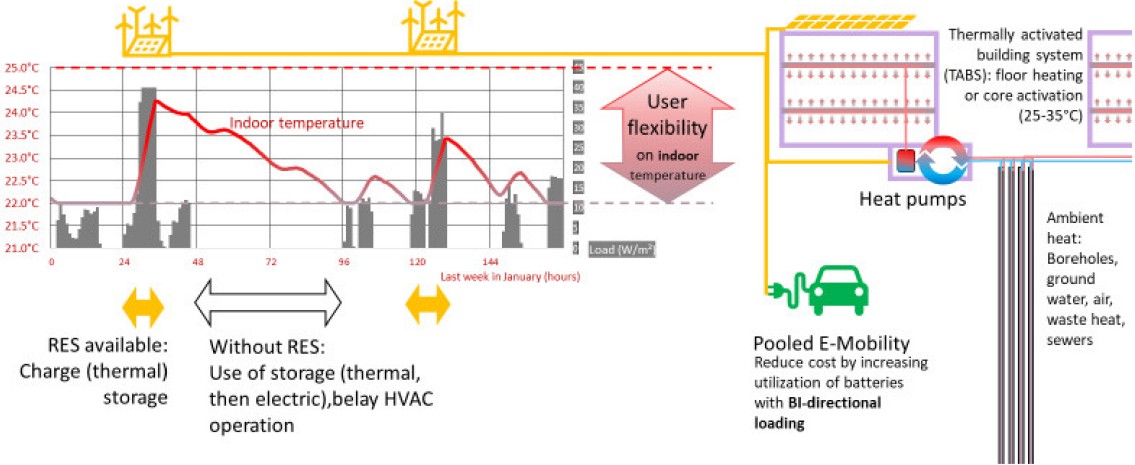

**Figure 3.** Example illustration of flexible DSM in a PED to maximize RES utilization.

## 6. Balance Targets (Are Reversed Context Factors)

By now it has become apparent that the typically assumed target of the energy balance, to be positive or above zero, needs further clarification and, as pointed out above, also justification. By rephrasing balance targets as contextual factors in a strictly positive balance, the targets can be parametrized to reflect any desired context. The balance target value becomes a target function: A key feature of the PED definition approach is that the target value is a variable quantity and does not have to be positive, per se. A variable target value for the balance across the system boundary can be considered as a virtual credit for the district balance. The quantitative target value is equivalent to an external credit/debit of the opposite sign:

$$\text{Primary Energy Balance} = \text{Weighted Exports} - \text{Weighted Imports} > \text{Balance target} \quad (1)$$

$$\text{Sum of Context factors} = \sum_i CF_i = -\text{Balance target (in a context)} \quad (2)$$

$$PED\ Balance = Weighted\ Exports - Weighted\ Imports + \sum_i CF_i > 0 \quad (3)$$

(1) is a specification of the more general description in [23], with a weighting of primary energy conversion and a general balance target. In (2), generic context factors are introduced in $CF_i$, denoting an arbitrary context factor, and the sum of these to be the negative of a balance target. (3) combines both (1) and (2), showing how the energy balance can maintain the nominal target > 0 while including context-sensitive targets.

Figure 4 illustrates and compares this approach with context factors in a situation, where the feasibility of the PED balance is accomplished by other means. It is an important consequence of this definition that any balance definition can be mapped to any other with an appropriate context factor. In other words, any inclusion or exclusion of energy services in the energy balance can be realized with the inclusion of an appropriate context factor. This can be leveraged to shift the discussion of system boundaries to a discussion of targets and appropriate context factors, which can offset balance components.

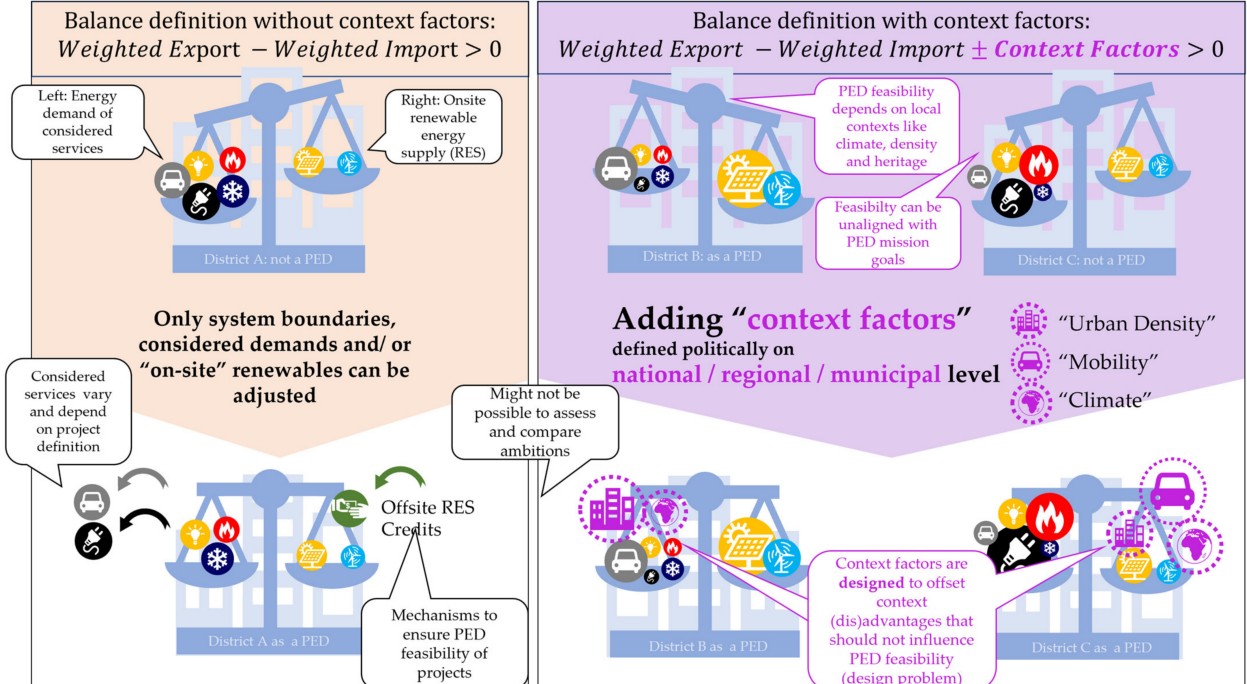

**Figure 4.** Illustration of the conceptual differences without (**left**) and with (**right**) explicit context factors.

Using this paradigm, a quantitative PED definition can be designed with a positive balance target using the context factors for the appropriate system boundaries depicted in Table 3. The inclusion of these ultimately virtual factors in the balance could be designed in an arbitrary number of ways. Their use must be rooted in its comprehensibility and link to the definition goals, which are examined in the following subsections, that can be quantified.

**Table 3.** Balance targets.

| System Boundary | Scope | Balance | Context Factors | Target | KPI |
|---|---|---|---|---|---|
| **Alpha** | Operation, use | Primary Energy Exports–Imports * | $\pm CF **$ Density | $> 0$ | kWh PE tot./$m^2$NFA/a |
| **Beta** | Operation, use, individual motorized mobility | Primary Energy Exports–Imports * | $\pm CF$ Density $\pm$ CF Mobility | $> 0$ | kWh PE tot./$m^2$NFA/a |
| **Omega** | Operation, mobility, and embodied emissions | GHG Emission Imports – Exports * | $-CF$ Emissions | $\leq 0$ | kg $CO_2$eq./$m^2$NFA/a |

* Energy flows into the district are counted negatively (e.g., grid electricity purchases and district heating, Table A1) and energy exports across the system boundary are accounted for positively (e.g., PV surpluses). The emission balance must be negative, the local and imported emissions are offset by export (external emission prevention) and a CF Emission, which represents an emission budget per reference area. ** CF—context factor.

### 6.1. PED Alpha Context: Density and the Feasibility for PED Districts in Urban Contexts

One of the main results of preliminary projects was to link the target value of the energy balance of a sustainable district to its building density, expressed by the floor area ratio (FAR). The approach was motivated by the observation that the energy balance depends significantly on the floor area ratio, i.e., the ratio between gross floor area (GFA) and plot size. This is because energy demands predominantly correlate with GFA, whereas potentials for local renewable energy generation generally are proportional to plot size [14]. The inverse proportionality between district density and achievable energy balance is both empirically evident [39] and analytically derivable. A definition that is to be comparably achievable in an urban context must therefore take this into account and refrain from a static target value. There are two aspects to consider here. First, this is simply a consequence of the physical facts, as detailed in [14,29,40]. However, there is a second point: in principle, the PED concept should not be opinionated about when and where it can be achievable and how easily. This is ultimately a political design question that can only be answered by quantitatively embedding the definition in the appropriate political context.

For Austria, this context is that the available building land reserves have been declining sharply, and, subsequently, land-use conflicts will become ever more prevalent due to geographic constraints [41]. Therefore, efficient land use has been a recurring political topic in recent years and has been called for with good reason. In Austria, therefore, special care must be taken to ensure that a PED definition does not encourage inefficient land use by making it easier to achieve in lower densities. Figure 5 illustrates the relationship graphically: the development on the left creates more usable space on the same lot and requires less infrastructure per person but would require infeasible amounts of PV (as the only commonly available source of local renewable electricity).

A 2021 study on the PED potential of urban typologies in Vienna, Austria found that only detached housing districts can achieve a positive annual energy balance (for heat and power) of 110%, whereas more dense typologies fail to achieve the criteria, with an annual balance ranking between 61% and 97% [40].

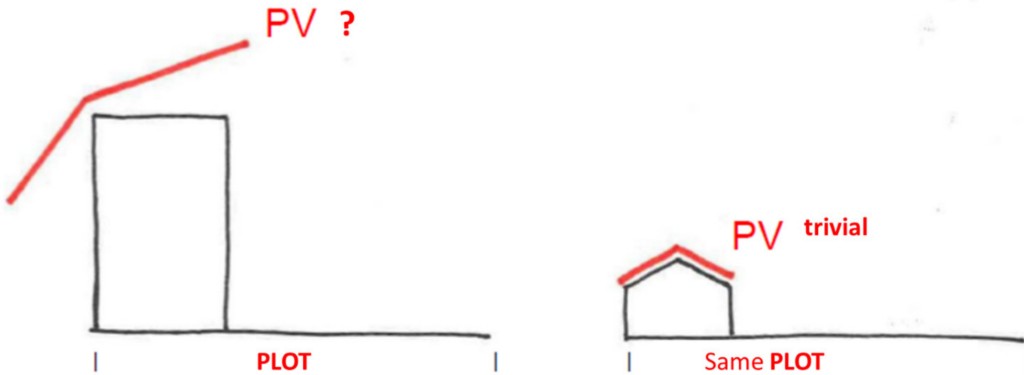

**Figure 5.** Illustration of the physical difference between high- and low-density districts in balancing their high and low energy use, respectively, with onsite renewables (represented by required PV area).

Figure 6 shows example energy balances of a number of districts in Austria that also show the impact of building density on the spread of the primary energy balance of districts. The vertical spread is caused by the different variants of a district (of constant density), from conventional variants without local renewable generation and efficiency measures at the lower end of the spectrum to "maximum" variants at the upper end, which feature highly efficient thermal hulls and HVAC systems with heat pumps for low-temperature heating and cooling and DHW heating and consider energy-flexible DSM and the maximum technical PV potential. The latter was determined on a project-specific basis, which inhibits a direct comparison between the maximum and minimum district variants. However, the representation of all analyzed district variants as one point on this FAR-to-PE balance diagram shows the connection with the building density and the target value derived from it. The red line represents the proposed context factor density as a modified target.

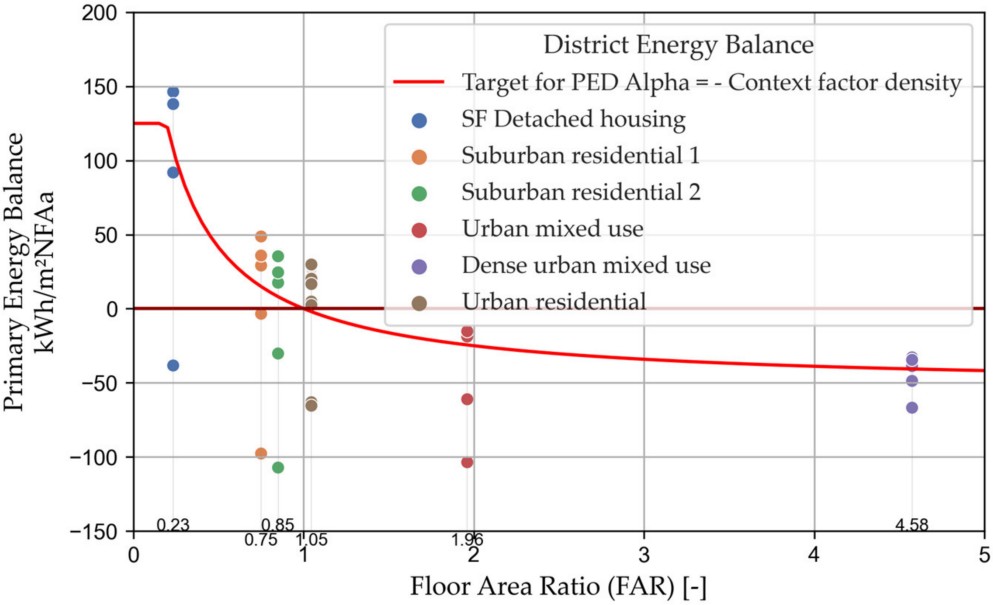

**Figure 6.** Primary energy balances of example districts over their density (FAR).

This correlation was also reported in similar studies [39] and comes as no surprise. The empirical relation between density and the possible PED balance can be also derived analytically. Based on a balance of energy demand and local renewable generation, the dependency on building density (FAR) is derived as follows:

$$\mathrm{BALANCE} = \mathrm{RES} - \mathrm{ED} \tag{4}$$

$$\text{BALANCE} = f_{\text{RES}} \text{ Plot Area} - f_{\text{ED}}\text{Floor Area} \tag{5}$$

with the onsite renewable energy supply (RES) and the local energy demand (ED), which both can be expressed as a product of the specific supply and demand per reference area ($f_{\text{RES}}$ per available plot area and $f_{\text{ED}}$ per useable floor area, see also Table 4). The division of the entire equation with the useable floor area changes the balance from absolute to floor-area-specific ($\text{kWh/m}^2\text{NFA}$) and reveals the analytical dependency on the floor area ratio, as it is defined as FAR = Floor Area/Plot Area:

$$\text{BALANCE(FAR)} = f_{\text{RES}}\frac{1}{\text{FAR}} - f_{\text{ED}} \left[\text{kWh}_{\text{PE}}/\text{m}^2_{\text{GFA}}\right] \tag{6}$$

**Table 4.** Formula abbreviations.

| Variable | Unit | |
|---|---|---|
| $\text{CF}_{\text{Type}}$ | $\frac{\text{kWh}_{\text{PE}}}{\text{m}^2_{\text{GFA}}}$ | Context factor of a given type (density, mobility) |
| $f_{RES}$ | $\frac{\text{kWh}_{\text{EEU}}}{\text{m}^2_{\text{PA}}}$ | Potential electricity yield per buildable plot area |
| $f_{ED}$ | $\frac{\text{kWh}_{\text{EEU}}}{\text{m}^2_{\text{GFA}}}$ | Electricity demand per gross floor area |
| $dx$ | $\frac{\text{m}^2_{\text{GFA}}}{\text{m}^2_{\text{PA}}}$ | Sensitivity parameter floor area ratio (FAR) |
| Cutoff | $\frac{\text{kWh}_{\text{PE}}}{\text{m}^2_{\text{GFA}}}$ | Maximum resulting energy balance |

Since the two parameters of the balance function $f_{\text{RES}}$ and $f_{\text{ED}}$ are conceptual variables, a scientifically deterministic determination is not expedient. Instead, they are considered control variables for the effort sharing between sparsely and densely built-up districts. They are determined politically within the framework of technical and economic feasibility according to the provision principle. For operational purposes, the formula for the density context factor was extended to include a cutoff and an offset in the *x*-axis (dx):

$$\text{Context Factor Density} = \text{CF}_{\text{Type}}(\text{FAR}) = -\text{Balance Target for PED Alpha Operation} = -\min \begin{cases} \frac{f_{\text{RES}}}{\text{FAR}+\text{dx}} - f_{\text{ED}} \\ \text{cutoff (max}value) \end{cases} \tag{7}$$

A comparison with Table 5 and Figure 7 shows that the factors were subsequently parameterized with approximately

$$f_{\text{RES}} \approx 30.4\frac{\text{kWh}_{\text{PE}}}{\text{m}^2_{\text{plot}}}, f_{\text{ED}} \approx 26.4\frac{\text{kWh}_{\text{PE}}}{\text{m}^2_{\text{GFA}}} \tag{8}$$

**Table 5.** Energy balance performance approximation of Austrian district typologies.

| District Type | Share | Potential Electricity Yield $f_{RES}$ | Electricity Demand $f_{ED}$ | $dx$ | Cutoff | Electricity Balance |
|---|---|---|---|---|---|---|
| | | $\frac{kWh_{EEU}}{m^2_{PA}}$ | $\frac{kWh_{EEU}}{m^2_{GFA}}$ | $\frac{m^2_{GFA}}{m^2_{PA}}$ | $\frac{kWh_{PE}}{m^2_{GFA}}$ | *PJ/a* |
| Unrefurbished | 0% | −1.0 | 50 | 0.15 | - | 0 |
| Thermal refurbishment | 40% | 0.0 | 38.5 | 0.15 | - | −42.16 |
| Refurbishment with minimal PV | 20% | 30.4 | 35 | 0.15 | 13 | −3.69 |
| PED Refurbishment | 20% | 30.4 | 26.4 | 0.15 | 62 | +11.55 |
| PED New construction | 20% | 30.4 | 26.4 | 0.15 | 62 | +11.55 |
| Total | | | | | | −22.7 |

It is important to note that the parameterization is not a quantitative representation of the technical potential of a district as a function of the FAR. In fact, the parameterization is motivated

by the technical potential, but the level of the factors is significantly lower than the actual technical potential and demand. Their level is derived from the resulting effort-sharing in the Austrian building sector. It is important to keep this in mind: the parameterization of the PED Alpha target through an equivalent context factor for "density" only operationalizes a small part of the technical potential difference. As a result, in the comparison of possible district configurations, some project variants are able to exceed the target value—sometimes significantly. These are mostly variants with technical but not economic feasibility.

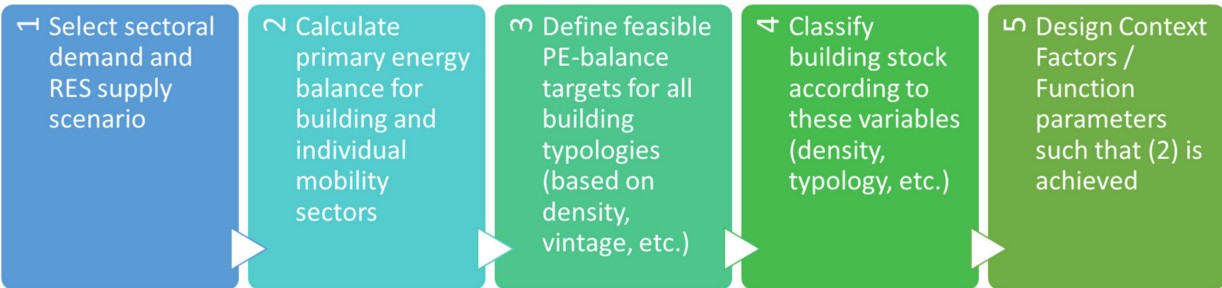

**Figure 7.** Parametrization steps of district density context factor.

The parameterization is carried out through the steps outlined in Figure 8. The target of the building sector is determined from the top-down consideration of the 100% renewable energy scenario Austria 2040 and the distribution according to the balance sheet allocation of generation and demand sectors presented at the beginning. With the assumption of the following refurbishment rates and classes, this results from Table 6 in the following parameterization across all FARs of the building sector:

$$\text{Balance Target Building sector} = \sum_{\text{Type}} \sum_{\text{FAR}=0}^{\infty} \text{CF}_{\text{Type}}(\text{FAR}) = -22.7 \, \text{PJ/a} \tag{9}$$

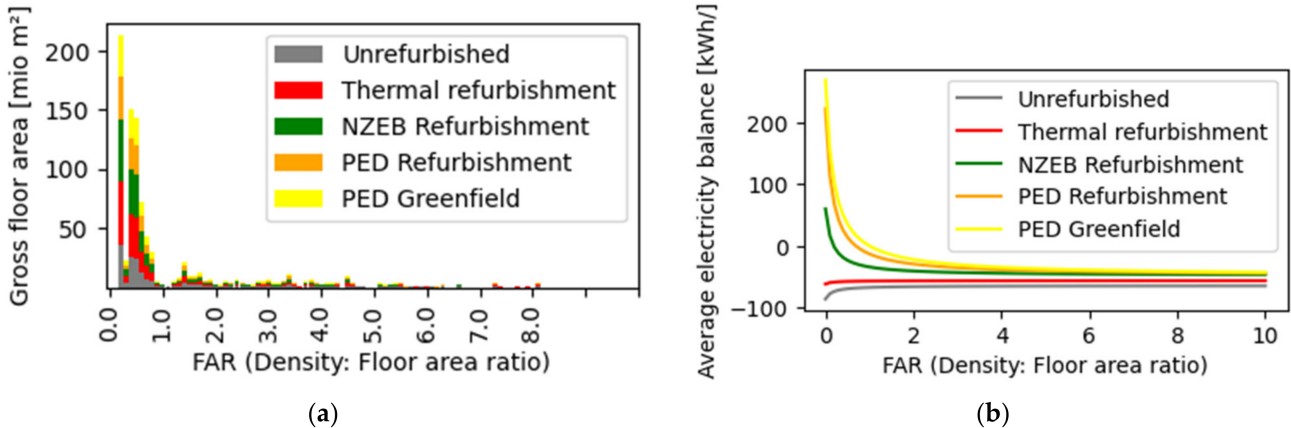

| (a) | (b) |

**Figure 8.** (**a**) Distribution of district typology gross floor area over the floor area ratio (FAR). (**b**) Distribution of district typology energy balance over the floor area ratio (FAR).

**Table 6.** Assumptions for the surrounding energy system.

| Energy Demands and Supplies 2040 | | Source |
|---|---|---|
| Electricity demand of the building sector 2040 for operation and MIT after sectoral allocation | 137.6 PJ/a | [24] based on [42] |
| Technical potential of the building sector energy demand 2030 | 48.2 PJ/a | [43] |
| **Allocation scenario** | | |
| Photovoltaics target 2040 (allocation to buildings) | 114.8 PJ/a | [24] based on [42] |
| Electricity balance target of the building sector | −22.7 PJ/a | |

As a next step, the share of PEDs in the building sector and the other district typologies are classified and their performance is estimated depending on density, as is shown in Table 5. Note that this is a first approximation and can and should be detailed as necessary to reflect the desired context.

Figure 8 shows the size and distribution of the defined district typologies depending on their density (a) and their respective energy balance target (b). This allocation can be tweaked to reflect other scenarios of building sector performance in a future energy system by changing the balance target of −22.7 PJ/a. It can also serve as grounds for discussion as to the relative performance and effort sharing of different parts of the building stock.

### 6.2. PED Beta Context: Mobility and the Surrounding Energy System

Nationally and internationally, there are conflicting opinions as to whether and in what form mobility should be included as an energy service in the system boundary of a PED. The main arguments are listed in the following Table 7.

The PED Beta system boundary is an attempt to expand the PED definition to include everyday mobility in the sense of the above-mentioned arguments without decreasing the feasibility of reaching a positive balance outright. For this purpose, the following approach was developed, which can be applied to all districts and neighborhoods in Austria at the beginning of project development without significant effort. Specifically, this is done by considering two additional components in the primary energy balance:

1. The mobility energy demand induced by the individual motorized mobility of the district as a statistical approximation. This is operationalized to depend on the public transport connection of the location, as well as the mix of uses in the district, which results in a district-specific mobility profile and the associated energy demand.
2. A project-extrinsic mobility energy budget, or context factor, from the surrounding renewable energy system, derived as the surplus from the regional renewable supply, which is allocated to the district via its share of useable floor space in the building sector.

**Table 7.** Arguments to include mobility in the PED energy balance.

| PRO Mobility Inclusion | AGAINST Mobility Inclusion |
| --- | --- |
| <ul><li>It corresponds to a more complete balancing of all energy and emission loads.</li><li>The quality of the location in terms of the everyday mobility induced by it can be assessed.</li><li>Concrete measures to reduce everyday mobility or the emissions caused by it, such as mobility sharing offers, charging infrastructure for e-cars, etc., should be quantitatively assessable.</li><li>Synergies of e-mobility charging infrastructure through the dynamic consideration of actual charging times and PV surpluses in the district and the advantages of energy-flexible districts can thus be mapped.</li></ul> | <ul><li>Lack of data and methods to reliably determine the energy balances and emissions of transport without great effort and uncertainty.</li><li>The plus-energy standard is made considerably more difficult or even impossible (in the sense of a strictly positive energy balance).</li><li>The scope of action for developers is significantly limited. Instead, it is primarily the municipality or city that must set specifications and, if necessary, take measures outside the building site.</li></ul> |

Together with the density context factor of PEQ Alpha, PEQ Beta thus fulfills a key requirement of this PED definition approach: the link to the surrounding renewable energy system and the mapping of effort sharing within Austria's building sector with the assessment target of a PED's energy balance.

Particularly in dense urban environments, it is questionable to mandate complete energy autonomy. Conversely, rural areas are more likely to have a surplus of renewables such as large wind and hydropower plants in close proximity. The design question is how these "regionally available renewables" can be allocated to and assigned to specific energy needs, such as in a district. Different approaches exist: the use of virtual system boundaries, which allows a form of offsetting (e.g., through the acquisition of credits); the distribution of the energy to districts through a provisioning quota (i.e., per inhabitant or floor area); or direct contracting with the operators of external RE plants.

All considerations ultimately lead to an allocation problem that should be openly addressed considering the wider energy system surrounding a district. Regionally available RES must be regionally balanced. Its availability must be allocated on a balance sheet to prevent individual actors from overusing the available resources, thereby creating a more difficult situation for the rest. The use

of available RES must be divided between the sectoral needs of industry and agriculture, mobility, and buildings through appropriate effort sharing. This has to happen on a superordinate level. Figure 9 shows an example allocation and resulting quantifiable provisioning of available RES that can be allocated to each district or building—not just PEDs. This "surplus" can be distributed per capita or floor area to the entire building sector with system boundary PEQ Beta, i.e., including individual motorized mobility. The quantification of this surplus is presented in [24].

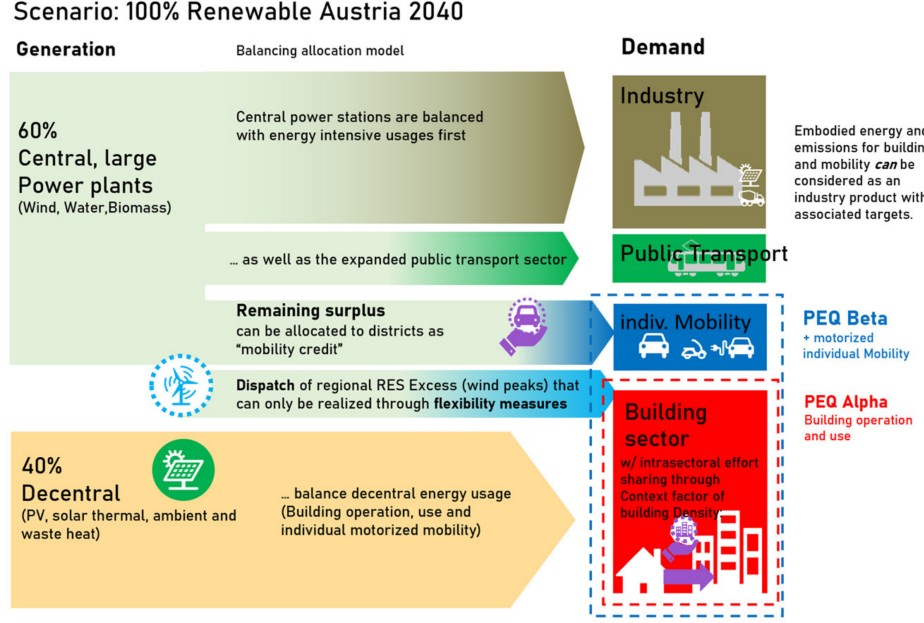

**Figure 9.** Schematic balancing of a national 100% renewable energy system 2040.

It is important to note that in this approach, no a priori allocation of external regional renewable energies is required for the building sector—and thus especially not for PEDs. This means that by and large, the building sector is responsible for its own renewable energy supply by the use of decentralized generation (e.g., PV, solar thermal, etc.).

### Derivation of the Mobility Context Factor of a PED

The integration of everyday mobility into the PED balance impedes its positivity potential unless there is a form of offsetting that does not have to be created directly at the site. We again propose to design a virtual context factor to fulfill this role. However, what mobility offsetting budget can be allocated to every district? We try to answer by looking at the district in the context of the Austrian national energy system "top-down". Assuming, according to current legislation and political aims, that Austria will be 100% renewable by 2040, a sectoral budget can be identified for projected energy supply and consumers that can be used per person to cover private everyday mobility. The following figure illustrates the national allocation as modeled in [24].

The surplus (or deficit) from large-scale renewable power plants is allocated to the entire resident population. In Austria, most scenarios of future 100% renewable energy supply result in a surplus from central power plants and thus a bonus for the individual districts. In other systems, there may just as well be a deficit, making the increased use of decentralized RES plants necessary to also account for more of its mobility energy demands. Assuming that these are also equally distributed to the building sector, this results in an additional malus or negative context factor. A PED Beta must then not only cover its local individual everyday mobility but also support the surrounding energy system with an additional local renewable surplus. However, the allocation illustrated in Figure 9 results in an electricity surplus from the regional "central large-scale power plants" of 6.3 TWh EE/a and a surplus from biomass of 0.36 TWhEE/a, or a total primary energy surplus of 10.7 TWhPE/a for use for MIV in the built environment, as shown in Table 8.

**Table 8.** Central surplus as budget to cover private everyday mobility in Austria per person.

| Large-Scale Renewable Power Plants | | |
|---|---|---|
| Electric (wind, hydropower) | 6.30 | *TWh/a* |
| Biomass | 0.36 | *TWh/a* |
| Primary Energy | 10.68 | *TWh PE$_{tot.}$/a* |

This surplus is distributed to all settlement and district areas in Austria on a pro rata basis according to the Austrian average share of this use in destination traffic, as shown in Table 9. About 50% of all Austrians' journeys home are for residential use. Accordingly, these journeys receive a 50% share of the total credit.

**Table 9.** Allocation of the central surplus budget by usage share of destination traffic and the distribution of usable NFA into a context factor per m²NFA.

| Usage | AT: Share of Destination Traffic [44] | Budget Share *TWh PE$_{tot.}$/a* | Usable NFA AT *mio m²$_{NFA}$* | CF* Mobility *kWh PE$_{tot.}$/m²$_{NFA}$/a* |
|---|---|---|---|---|
| Residential | 50% | 5.30 | 375.6 | 14.11 |
| Office and commercial | 21% | 2.20 | 53.1 | 41.47 |
| Education | 3% | 0.29 | 22.5 | 12.93 |
| Retail and other | 27% | 2.89 | 96.5 | 29.97 |
| **Total** | 100% | 10.68 | 547.8 | |

* CF: context factor.

At the same time, the energy demand of motorized individual mobility must be determined by means of an Austria-wide statistical allocation of usable areas to inhabitants and their average annual trip kilometers, as outlined in [25,44] and shown in Figure 10 for a range from a hundred percent EV use in usage color to a hundred percent fossil vehicles in gray. It shows that, on average, the budget—or mobility context factor—in blue is not sufficient to cover the energy demand of private everyday mobility for most locations in Austria. In general, PED Beta is thus more difficult to achieve than PED Alpha and more ambitious. In particular, however, the energy demand depends on the remaining share of private transport via the location-dependent public transport quality and can deviate by up to 50% upwards and downwards from the Austrian mean value. This allocation has the effect that in "rural" areas with poor public transport accessibility, the target value is more difficult to achieve than in "urban" areas with higher public transport density.

### 6.3. Context Outlook: Emission Budgets and Embodied Emission Context

The PED Omega system boundary represents the final shell of this methodology. It aims to enable to balance of the entire climate change-related environmental impacts of a district and make them comparable with a target value or context-sensitive budget. This enables a statement on whether the district is compatible with the demands of a future climate-friendly, emission-neutral society or to which extent it would require further measures and possibly retrofitting down the line. Despite or precisely because of the methodological complexity, it is now more necessary than ever to start with the quantitative linking of (inter)national and individual climate goals of each person and to locate them and make them visible where there is also concrete scope for action for this—as in the case of district planning—and where it is possible to set the course for achieving the 2040 climate goals. The context factor or credit for PED Omega is currently under development and consists of three parts: Firstly, a basic personal credit that represents the emission target of 800 kg $CO_2$equiv/Pers/a, which represents the share that is allocated for the operation of the building and its production and repair. In addition, it includes two credits, which are calculated analogously to the first two system boundaries PED Alpha and PED Beta and result from a conversion of the primary energy credits for building density and a mobility budget into GHG emission equivalents. The detailed methods and parametrizations will be published once finalized and tested.

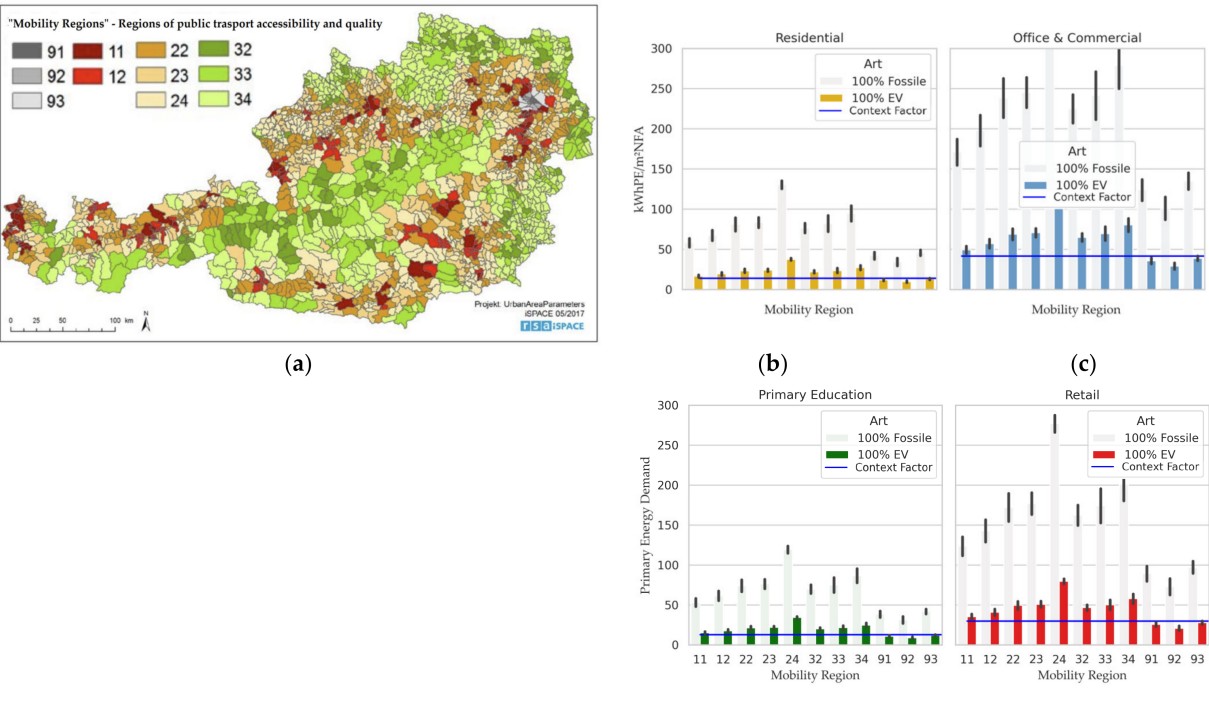

**Figure 10.** (**a**) AT "Mobility Regions" [44] (**b**–**e**) Energy demand for motorized individual mobility (bars) in these mobility regions for 100% fossil (grey), 100% EV shares (color), and offsetting context factor (blue line). (**b**) Residential (orange). (**c**) Office and commercial (blue). (**d**) Primary education (green). (**e**) Retail (red).

## 7. Definition Application Examples

This section gives a brief overview of the example assessments of seven different Austrian district typologies. As shown in Table 10, these districts represent different usage patterns and cover a wide range of different densities from suburban and rural detached single-family homes to dense urban typologies of different uses with up to seven stories. The definition was designed to be feasible for all typologies with extensive additional measures of onsite RES, energy-flexible operation, and partial electrification and reduction in private motorized mobility. The resulting energy balances are shown in Figure 11. All points represent a district configuration with energy demand on the *x*-axis and renewable energy supply on the *y*-axis. The points above the dashed 45° line are considered PEDs, following the established convention of supply exceeding demand. Districts show the following configurations, which each add to the last: (1) baseline (in grey, project-specific, varying ambition); (2) with additional measures (in orange, maximum ambition of the project); (3) with consideration of the density context factor (in red)—note that it adds both virtual demand (horizontally) or supply (vertically) depending on the district density (FAR); (4) included energy for individual motorized mobility (in black, varying share of EV, around 50% of all car trips); and (5) including the context factor designed to offset the mobility energy in the balance (in blue).

Note how the positivity of the project balance without context factors correlates with density: higher FAR districts cannot achieve a positive balance even with the most ambitious measures, whereas districts of lower density could (over)achieve a positive balance, the lower the density the easier, without ambitious measures.

**Table 10.** Overview of example districts.

| District | NFA $m^2$ | FAR | Type | Usage |
|---|---|---|---|---|
| Dense Urban Education | 25,009 | 5.80 | Refurbishment | |
| Dense Urban Mixed-Use | 26,805 | 4.58 | Green field | |
| **Urban Mixed-Use** | 43,778 | 1.96 | Green field | |
| **Urban Residential** | 40,383 | 1.05 | Green field | |
| **Suburban Residential 1** | 19,838 | 0.85 | Green field | |
| **Suburban Residential 2** | 4098 | 0.75 | Green field | |
| **SF Detached housing** | 124 | 0.23 | Greenfield | |

Usage bar chart legend: Residential, Office & Commercial, Secondary Education, Primary Education, Retail. Values: 50% / 50%; 52% / 38% / 8%; 41% / 54% / 5%; 97% / 2%; 91% / 4%; 100%; 100%.

Suburban Residential 1 — Urban Residential — Suburban Residential 2 — Urban Mixed-Use — Dense Urban Mixed-Use

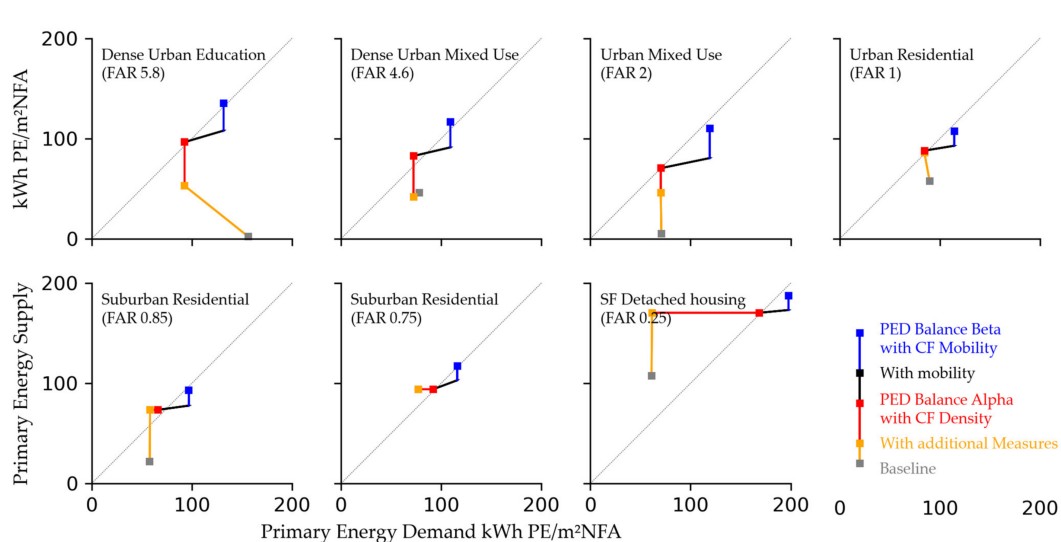

**Figure 11.** Primary energy balance of seven example districts.

## 8. Discussion

### 8.1. Context Factors Compared to Other Offsetting Mechanisms Such as RES Credits

The main difference is not in its effect but rather in its conceptual perspective and its resulting comparability. A context factor is just a framework to include an arbitrary design goal into the quantitative definition and formulate it as explicitly as possible. In fact, practically anything can be mapped into a context function and, as such, included in a given quantitative framework. Arguably, "context function" might be a more appropriate term, as it almost always is also dependent on the project's internal and external variables.

This may even proof useful for the comparison of different quantitative definitions themselves. The resulting balance for a district under a certain assessment can be reinterpreted as its own context function and compared. With enough such PED balance assessments, it could even be useful to compare the effects of different system boundaries and weighting systems in this manner. This also means that, in contrast to, e.g., [4] (p.13), we see the framework of context factors capable of flexibly

encapsulating enough contexts of different districts, so that the other parts of the balance definition, i.e., the system boundaries and weighting systems, can omit their current task of modulating feasibility and instead be defined uniformly and comparatively.

### 8.2. Why Is the Context Factor for Mobility Allocated per Usable NFA and Not per Person?

The advantage of this approach is that neither the resulting credit nor the mobility energy demand induced by the district depends on the actual occupancy density and number of users in the district. The approach is, therefore, more uniform and not dependent on the actual density of people in the district, which proves difficult to determine in practice.

### 8.3. Why Is Mobility Only Taken to Include Individual Motorized Mobility?

A guiding principle of the PED definition is that of subsidiarity, according to which the smallest possible unit in a system should have the greatest possible autonomy in dealing with tasks. A district can affect individual forms of mobility through its location, design, and measures, which is why it should be made partly responsible in the context of a PED definition. Public transport, on the other hand, is a superordinate mode of transport that primarily serves to connect districts with each other and with other means of transport. The accessibility of a district or settlement can strongly depend on the available public transport infrastructure; conversely, district developers cannot necessarily, and actually only in exceptional cases, influence the design of public transport. Moreover, more energy-intensive infrastructure is necessary for public transport, which also serves the general public beyond the district. For these reasons, the energy and emission provisions for public transport should be situated at a higher than district level. These provisions could, in turn, be allocated to the district, which then gives rise to the non-trivial question of the allocation method, which is even more complex for public transport than for individual everyday mobility, because the occupancy density and the allocation of trips to floor usage are less clear.

The district development timeline also plays a crucial role here, as district projects and their development with public transport do not always take place at the same time and in accordance with the initial plan.

### 8.4. Consideration of Delivery and Other Occupational Traffic

Delivery and occupational traffic can be considered, given an appropriate dataset, which was not available in this definition. Namely, if the underlying annual transport survey includes occupational and delivery journeys differentiated by targeted space use, i.e., home deliveries and occupational deliveries. However, again, inclusion will likely only serve the PED definition design goals if district measures can be quantified and linked to a derivative target from the surrounding.

### 8.5. Non-Everyday Mobility and Air Traffic

This is excluded from consideration for similar reasons: There is no methodology to assess the effect of district locations and measures on non-commonplace mobility. On top of that, it is predominantly influenced by individual lifestyles and it is not clear if and how energy and emission targets can be derived here.

### 8.6. Hourly Weighting of Energy Flows

Instead of monthly weighting, which improves the accuracy compared to annual primary energy conversion factors, it is considered to use hourly conversion factors. Providers such as ElectricityMap [45] already provide both historical and real-time data for emission intensities. Conversion to primary energy content is in principle possible but has not yet been standardized. In addition, forecasts of hourly PE and GHG intensities are of particular interest in the case of corresponding demand and generation developments.

### 8.7. Timeframe: Current, Future, or Cumulative?

Based on the difference in primary energy conversion today and in the future, the question of different observation periods arises. What statements and consequences do these different considerations and standards have on the measures and projects derived from them? What are the advantages and disadvantages and how can they be combined? In view of the advancing climate catastrophe, not only the statistically annual but also the cumulative consideration of emissions up to 2040, taking into account the time of emission, is relevant, even if this would pose additional challenges for operationalization.

## 8.8. Existing Districts and Refurbishment

The renovation sector was largely excluded from the analysis, although it will of course play the most important role in the coming years. In principle, the definition and operationalization presented here can also be applied to the refurbishment of existing buildings and, as initial studies show, can sometimes be achieved. However, it is clear that especially the PED Alpha system boundary with the relatively high implicit requirements for energy efficiency and local renewable energy production will not be easily—if at all—achievable for all existing quarters. Here, apart from the building density, the credit must also be examined and, if necessary, parameterized depending on additional parameters such as the building age or the settlement typology.

Although the presented definition can be used for both green and brown field development, there currently is no specific context factor considering potentially lower balance feasibility for refurbishments. As the first two boundaries, Alpha and Beta, do not include embodied emissions, they are easier to achieve for green field developments, whereas the opposite is true for brown field and refurbishments, where the embodied emissions due to materials are lower and thus favor the last definition, Omega. It is unclear as of yet if this distinction is sufficient in theory and practice, as the sample size of refurbishment PEDs is still small and only now expected to expand with the investigation of the second round of JPI UE PED projects.

## 8.9. Data Availability and Accessibility

One potential challenge of the presented approach is that of data availability and accessibility, as it partly relies on data that are not readily available for all buildings and districts in Austria, especially in brown field developments. Amongst these, the most critical are (1) hourly data on external grid flexibility requirements and normative methods to obtain them, (2) time-sensitive primary energy conversion weighting factors in general, and (3) mobility data. Further research must yield possible data sources, and normative standardization processes must formalize a standardized dataset for certification.

## 8.10. Possible Implications of the Proposed Definition in Shaping Future Policies

As one of the design goals of the definition is to lead to a national PED certification, it is important to reflect on the possible impacts and implications of the proposed definition. First, the density context of the definition shifts the ambition pressure to the side of low-density developments, which might be a position not justifiable by regulation and legislation. Second, the introduction of a certifiable PED definition with a purely technical character might set wrong incentives for district developments to forego other certifications that have more emphasis on social and ecological assessments that should be used in conjunction with the proposed definition. Third, the rigidity, ambition, and complexity of the framework might deter potential PED districts from pursuing such a standard.

## 9. Conclusions

PED definition must be understood as a design problem and cannot be detached from the goals and aims to be furthered by it. Indeed, it is only these goals and aims that make the definition useful. The positive energy balance is the unique feature of the PED concept, and these three design choices on boundaries, weighting, and targeting can be used to define PEDs and assess some aspects of energy efficiency, RES, and even flexibility together within the energy balance. Using only a single indicator also lends itself to standardization and certification. Crucially, the inclusion of dynamic balance targets allows a comprehensive link between balance assessment and definition goals. However, such a balance cannot be used to assess all relevant aspects of PED development by itself and needs to be accompanied and complemented by other assessment and certification systems, be it on a regulatory or project level.

In line with its PED definition design goals, this paper introduced a theoretical framework of a quantitative energy balance that can be defined with relative uniformity while still leaving room for contextual targeting. With this framework, it is possible to contextualize PED balance assessments on different frames of regionality, from international to municipal. The paper shows that setting PED balance targets is analogous to quantifying context impacts, which allows for a formalization of PED contexts and can improve comparability between projects and other quantitative balance definitions. This general framework was used to construct three distinct PED definitions that are needed for PED planning and certification practice in Austria: PED Alpha for operation and use, PED Beta for included private mobility, and PED Omega for embodied emissions from construction and mobility.

The effects of this PED definition design were exemplified by seven district assessments that show its feasibility in various green field contexts. The use of a context factor for density allows both district typologies of very high (FAR > 3) and low (FAR < 1) density to achieve a positive energy balance for operation with comparable ambition for energy efficiency, energy flexibility, and onsite renewable generation measures. Furthermore, the use of a context factor for mobility derived as a credit from the surrounding energy system allocated by usable floor area can be used to offset the energy demand induced by including individual motorized mobility in the energy balance. With this, it is feasible for both urban and rural districts and neighborhoods to achieve a positive energy balance with, again, similar ambition in terms of reduced motorized mobility demand and a switch to electric vehicles.

**Author Contributions:** Conceptualization, S.S., methodology, S.S. and T.Z.; validation, S.S., D.S., and T.Z.; investigation, S.S.; resources, S.S. and D.S.; writing—original draft preparation, S.S.; writing—review and editing, J.B.; supervision, J.B. All authors have read and agreed to the published version of the manuscript.

**Funding:** This research was partly funded by the Austrian Research Promotion Agency (FFG) research project "Future District Austria" (German) (online: https://nachhaltigwirtschaften.at/de/sdz/projekte/zukunftsquartier-oesterreich.php (accessed on 30 March 2023).

**Data Availability Statement:** Publicly available datasets were analyzed in this study. This data can be found here: [https://github.com/simonschaluppe/peexcel, accessed on 30 March 2023].

**Conflicts of Interest:** The authors declare no conflict of interest.

**Abbreviations**

| | |
|---|---|
| AT | Austria |
| CF | Context factor |
| DHW | Domestic hot water |
| DSM | Demand-Side Management |
| EU | European Union |
| GHG | Greenhouse gas |
| GFA | Gross floor area |
| JPI UE | Joint Programme Initiative Urban Europe |
| KPI | Key Performance Indicator |
| MIT | Motorized individual transport |
| PA | Area of buildable plot or parcel |

**Appendix A**

**Table A1.** Grid electricity conversion factors.

| Month | Primary Energy kWh/kWh | GHG-Emissions kgCO$_2$eq./kWh |
|---|---|---|
| January | 1.80 | 0.304 |
| February | 1.79 | 0.304 |
| March | 1.72 | 0.264 |
| April | 1.58 | 0.211 |
| May | 1.47 | 0.167 |
| June | 1.46 | 0.163 |
| July | 1.44 | 0.163 |
| August | 1.48 | 0.167 |
| September | 1.58 | 0.208 |
| October | 1.71 | 0.260 |
| November | 1.77 | 0.282 |
| December | 1.79 | 0.291 |
| **Average** | 1.63 | 0.231 |

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
