# Peer review of "A Quantitative Positive Energy District Definition with Contextual Targets"

_buildings, doi:10.3390/buildings13051210_

Round 1
Reviewer 1 Report
The title of research paper is interesting and it is explored very well. Note down the below review points and address it accordingly.
1. Try to avoid giving short term like PED in title, What is PED (try to give full term for PED, in title)
2. In abstract, try to give full form for PED and RES
3, Need to provide few more keywords in Abstract
4. In page2, write short name with full form (like GHG, KPI, IEA, EPBD, and EU)
5. In Section 3, goals of PED definitions, in that, how the proposed research work, will be helpful to fulfil the goals, try to give justification of statement, on what way, the proposed study will be helpful to achieve the mentioned goals in section 3
6. In page 5, write full form for LEED, BREAM
7. Most of the Figure name is too long with explanation, try to shorten the name of figure with clear content
8. Section 5.1, the sub heading name and short name DSM is not matching
9. Section 5.1, try to shorten the contents and write clearly, its too lengthy explanations
10. Equations 1,2, and 3 need to be acknowledged with appropriate sources of references
11. need to provide abbreviations for the variables used in equations 6, 7,8, and 9
12. In Table 4, some heading names for the contents inside tables are missing (only showed "source")
13. Figure 9, names to be reassigned as 9a and 9b, respectively
14. Table 7, large scale renewable power plants, to be aligned with right hand side table of contents
15. need to assign, Figure 11, numbering individually, 11a to 11e and the contents of figures 11 (x and y scale factors) is not clearly visible
16. Conclusions part needs to be shortened and need to write the outcomes of proposed study more clear in related to the objectives of PED
Need to improve write up of English language
Author Response
Thank you very much for your detailed review. We have tried addressed your feedback in full with the following actions:
- Try to avoid giving short term like PED in title, What is PED (try to give full term for PED, in title)
- Done!
- In abstract, try to give full form for PED and RES
- Done!
3, Need to provide few more keywords in Abstract
- Was: Keywords: Positive Energy District, PED definition, context factors, PED assessment,
- Added: Energy Transition, Energy balance assessment, Sustainable Districts, Key Performance Indicators
- In page2, write short name with full form (like GHG, KPI, IEA, EPBD, and EU)
- Done!
- In Section 3, goals of PED definitions, in that, how the proposed research work, will be helpful to fulfil the goals, try to give justification of statement, on what way, the proposed study will be helpful to achieve the mentioned goals in section 3
- Added the following justification on the connection between paper and goals:
- "In this paper, The introduction of a quantitative PED definition assessment scheme using context-factors is aimed to facilitate these goals. In particular, the density context-factor addresses Goals 1-3 and 5 by connecting national PV capacity targets (Goal 1 and 5), depending on density (Goal 2) and is comparably achievable for different usages (goal 3) as shown in section _ . Goal 4 is facilitated by introducing other means of achieving a nominally positive energy balance. The proposed is also compatible with the JPI UE Framework definition (Goal 6), as the latter also mandates a positive energy balance, but also expects some form of contextualization. This paper also tries to further goal 7 by intrudcing a formal definition framework that can subsequently be parametrized for national standardization and certification. Goal 8 is addressed by the differentiation of the definition into 3 possible perspectives through the introduction of 3 possible system boundary (Alpha, Beta, Omega)"
- In page 5, write full form for LEED, BREAM
- Done!
- Most of the Figure name is too long with explanation, try to shorten the name of figure with clear content
- Shortened Caption 2 to: The three parts of a this PED definition via an energy balance assessment
- Caption 3 shortened to: Types and Extent of defined system boundaries
- Caption 4 shortened to: Example illustration of flexible DSM in a PED to maximize RES utilization.
- caption fig.7 shortened to: Primary energy balances of example districts over their density (FAR).
- Caption Figure 6 shortened to: Illustration of the physical difference between high- and low-density districts in bal-ancing their high and low energy use respectively with onsite renewables (represented by re-quired PV area).
- Caption Figure 10 shortened to: Schematic balancing of a national 100% renewable energy system 2040.
- Capion Figure 11 shortened.
- Section 5.1, the sub heading name and short name DSM is not matching
- subheading changed to "5.1 Energy-flexible grid use through demand side management (DSM)"
- Section 5.1, try to shorten the contents and write clearly, its too lengthy explanations
- section rewritten and shortened from 5000 to 2500 letters
- Equations 1,2, and 3 need to be acknowledged with appropriate sources of references
- added reference to sartori, 2014
- referenced formulas in text
- need to provide abbreviations for the variables used in equations 6, 7,8, and 9
- added table with formula abbreviations
- In Table 4, some heading names for the contents inside tables are missing (only showed "source")
- Added missing headings
- Figure 9, names to be reassigned as 9a and 9b, respectively
- Done
- Table 7, large scale renewable power plants, to be aligned with right hand side table of contents
- Improved alignment
- split table captions into two
- need to assign, Figure 11, numbering individually, 11a to 11e and the contents of figures 11 (x and y scale factors) is not clearly visible
- split figure into Fig11.a-e
- improved readability
- Conclusions part needs to be shortened and need to write the outcomes of proposed study more clear in related to the objectives of PED
- Conclusion was shortened and rewritten to address the outcomes of the study more clearly in terms of the PED objectives

Reviewer 2 Report
Overall, the paper presents a comprehensive framework for defining and assessing PEDs through a quantitative energy balance approach. The goals and aims of PEDs are clearly identified as crucial components of the definition process, and the paper provides a clear link between balance assessment and definition goals through the use of dynamic balance targets. The inclusion of dynamic balance targets also allows for a formalization of PED contexts, improving comparability between projects and other quantitative balance definitions.
The use of a context factor for density and mobility allows for both urban and rural districts to achieve a positive energy balance with similar ambition in terms of energy efficiency, flexibility, and onsite renewable generation measures. The feasibility of the framework is demonstrated through seven district assessments in various green field contexts.
However, the paper would benefit from further elaboration on the potential limitations or challenges that may arise in implementing this framework in practice, particularly in terms of data availability and accessibility. Additionally, it would be useful to provide a discussion on the potential implications and impact of the proposed PED definition and assessment framework in shaping future building and energy policies.
Simplification of complex issues: The authors argue that all three pillars of PEDs (energy efficiency, onsite renewables, and energy flexibility) can be assessed with a single metric of primary energy balance. However, this oversimplifies the complex interactions between these different factors and may not accurately capture the full scope of PEDs.
Lack of consideration for social and economic factors: The paper focuses primarily on technical aspects of defining PEDs and does not consider the social and economic factors that may impact energy use and emissions in a given district.
Author Response
Thank you very much for your review!. We tried to address your feedback with the following actions:
Review: However, the paper would benefit from further elaboration on the potential limitations or challenges that may arise in implementing this framework in practice, particularly in terms of data availability and accessibility. Additionally, it would be useful to provide a discussion on the potential implications and impact of the proposed PED definition and assessment framework in shaping future building and energy policies.
- Added Discussion: Data availability and accessibilty:
- "
One potential challenge of the presented approach is that of data availability and accessibility, as it partly relies on data that is not readily available for all buildings and districts in Austria, especially in brown field developments. Amongst these, the most critical are 1) hourly data on external grid flexibility requirements and normative methods to obtain them, 2) time-sensitive primary energy conversion weighting factors in general, 3) mobility data. Further research must yield possible data sources, and normative standardization processes must formalize a standardized dataset for certification.
"
- Added Discussion: "
- Potential implications and impact of the propsed PED definition in shaping future building and energy policies
As one of the design goals of the definition to lead to a national PED certification, it is important to reflect on possible impacts and implications of the proposed definition. First, the density context of the definition shifts the ambition pressure to the side of low-density developments, which might be a position not justifiable by regulation and legislation. Second, the introduction of a certifiable PED definition with a purely technical character might set wrong incentives for district developments to forego other certifications that have more emphasis on social and ecological assessments, that should be used in conjunction with the proposed definition. Third, the rigidity and ambition of the framework might deter potential PED districts to pursue such a standard. "
Simplification of complex issues: The authors argue that all three pillars of PEDs (energy efficiency, onsite renewables, and energy flexibility) can be assessed with a single metric of primary energy balance. However, this oversimplifies the complex interactions between these different factors and may not accurately capture the full scope of PEDs.
- Conculsion: changed to: "Some aspects of energy efficiency, RES and even flexibility can be assessed together within the energy balacne. this alone cannot capture all the indicators to accurately assess the various PED design goals, but it could make standardization and certification easier, in terms of a common denominator, that still encapsulates some, of not all aspects, and would need to be accompanied and compelemented by other assessment systems, that might be selected project specifically.
Lack of consideration for social and economic factors: The paper focuses primarily on technical aspects of defining PEDs and does not consider the social and economic factors that may impact energy use and emissions in a given district.
- Reaction: True! The lack of consideration is argued in Section 3.1, lines 200-207.
- It was now also adressed in the Conclusions as a limitation to the presented approach. lines 721 ff.

Round 2
Reviewer 2 Report
Most of the issues have been resolved and the current version is ready for publication under editorial consideration.